# A 30-Year Probability Map for Oil Spill Trajectories in the Barents Sea to Assess Potential Environmental and Socio-Economic Threats

Victor Pavlov [1,*], Victor Cesar Martins de Aguiar [2], Lars Robert Hole [3] and Eva Pongrácz [1]

[1] Water, Energy and Environmental Engineering Research Unit, University of Oulu, P.O. Box 4300, FI-90014 Oulu, Finland; eva.pongracz@oulu.fi
[2] Department of Physics and Technology, UiT The Arctic University of Norway, N-9037 Tromso, Norway; victor.d.aguiar@uit.no
[3] Oceanography and Marine Meteorology, Norwegian Meteorological Institute, P.O. Box 7800, N-5020 Bergen, Norway; lrh@met.no
[*] Correspondence: victor.pavlov@oulu.fi

**Abstract:** Increasing exploration and exploitation activity in the Arctic Ocean has intensified maritime traffic in the Barents Sea. Due to the sparse population and insufficient oil spill response infrastructure on the extensive Barents Sea shoreline, it is necessary to address the possibility of offshore accidents and study hazards to the local environment and its resources. Simulations of surface oil spills were conducted in south-east of the Barents Sea to identify oil pollution trajectories. The objective of this research was to focus on one geographical location, which lies along popular maritime routes and also borders with sensitive ecological marine and terrestrial areas. As a sample of traditional heavy bunker oil, IFO-180LS (2014) was selected for the study of oil spills and used for the 30-year simulations. The second oil case was medium oil type: Volve (2006)—to give a broader picture for oil spill accident scenarios. Simulations for four annual seasons were run with the open source OpenDrift modelling tool using oceanographic and atmospheric data from the period of 1988–2018. The modelling produced a 30-year probability map, which was overlapped with environmental data of the area to discuss likely impacts to local marine ecosystems, applicable oil spill response tools and favourable shipping seasons. Based on available data regarding the environmental and socio-economic baselines of the studied region, we recommend to address potential threats to marine resources and local communities in more detail in a separate study.

**Keywords:** Arctic marine resources; oil drift; OpenDrift; OpenOil; oil spill simulations





## 1. Introduction

Oil spill response research commenced after the Torrey Canyon spill in 1967, the first major supertanker disaster [1]. Even though the yearly number and severity of spills have since reduced, oil spills are still a recognised problem worldwide [2]. Lately, a lot of attention focuses on the Arctic [3], which has thus far been spared of major incidents. There is a variety of technological oil spill response applicable solutions, but they all have certain operational limitations due to regional environmental challenges [4]. As a result of rising oil exploration and industrial activities in the circumpolar seas (e.g., Norwegian Sea, Barents Sea, Kara Sea), the risk of oil spill accidents is much higher than before [5–8]. The situation is also aggravated by the Arctic ice cap melting and the resulting increase in maritime traffic in the area [9,10]. In light of the projected increase of shipping activities and the geographical scarcity of response centres, here, we address the following research questions: how likely is it that an oil spill would reach the sensitive areas; what are seasonal and spatial variations, and how applicable can oil spill response tools be for the local conditions? The objective of this research is to address potential threats to marine resources and local communities from oil spill accidents. More specifically, this paper aims at defining potential

hazards to the local environment based on the simulated accidents of two fuel oil types. Our approach takes into account oil mass weathering during spreading, emulsification, vertical mixing, natural dispersion by wave breaking and evaporation as well as being based on high resolution ocean models. Oil weathering and drift over 30 years from continuous oil spills at one given location along popular maritime routes are calculated.

The focus of this study is on the Barents Sea of the Arctic Ocean. It accommodates popular maritime traffic lines, which then join into the Northern Sea Route. Despite its recognised high economic and biological value and availability of various environmental data sources, published results are scarce regarding simulating and analysis of offshore oil spill emergencies in the Barents Sea [11,12].

There are a number of tools to simulate oil spill accidents globally (e.g., GNOME, OSCAR, Seatrack Web, Oilmap, MEDSLIK, as well as 12 other models) and predict their behaviour and fate in water [13]. Many of them are commercial access only and have some additional functions. The probability map for this study was generated in OpenDrift due to its broad functionality, open access and ease of use.

Similar studies were conducted in the Eastern Mediterranean Sea. The approach was based on another open-source code Lagrangian oil spill model—MEDSLIK. Shoreline susceptibility analysis was conducted using Environmental Susceptibility Index (ESI) which was defined in Adler and Inbar [14–17]. Our work also includes reference to a sensitivity study of the Barents Sea region but it is based on offshore biological distribution of local species instead of bathymetric and geomorphological data and ESI index allocation for the shoreline. Both OpenDrift and MEDSLIK represent comprehensive operational oil spill models, which are most widely-used by researchers due to their input data requirements and characteristics [13].

## 2. Resources of the Barents Sea

### 2.1. Environment

The Barents Sea, being part of the Arctic Ocean, is positioned between 81° N and 66° N latitudes and 16° E and 68° E longitudes, and has the mean depth of 350 m, with the total area of 1,405,000 km$^2$. The sea is situated above the Polar Circle, where periods of darkness may last annually up to 4 months. The southern border of the sea is drawn by the Eurasian continent, as illustrated in Figure 1. All seas of the Arctic Ocean are young in geological terms and have the same origin. In times of the ice age, there was a large land area covered with ice instead of the ocean. After the ice age era, all lowlands were submerged in water, whereas highlands remained, turning into islands and peninsulas [18,19]. A large part of the Arctic Ocean, especially its periphery, is shallow—about 200 m in depth. There are many islands in the sea area, the coastline is sophisticated with fjords and bays as well as being mostly rocky and steep [20–22].

The Atlantic Ocean current coming from the west through the Norwegian Sea has an influence on the sea in terms of salinity, temperature and wind. The water is warmer, compared, for example, to the neighbouring on the east of the Kara Sea, which has low temperatures due to its central location in the Arctic Ocean. The water temperature of the Barents Sea ranges from 0 °C in winter to 9 °C in summer. The stratification in the water column is almost absent; instead, there is a characteristic feature of intensive water mass mixing in the sea. The salt content is about 35 g of salt per water kilogram [23]. Due to warm water masses brought from the Atlantic, the south-east of the sea never becomes frozen, and oil spill response (OSR) operations differentiate essentially for the given conditions. The Barents Sea, for example, in comparison with the Kara Sea, is biologically rich and productive: also due to active interaction with the Atlantic Ocean. During spring bloom, March to June, there may be primary production of phytoplankton of up to 400 mg m$^{-3}$ day$^{-1}$. During summer, the upper 30 m of water column are abundant with it. Hence, there is also a large presence of zooplankton—up to 500 mg m$^{-3}$ [24]. Even though the sea is relatively pure historically from chemical contamination, its ecosystems

are less metabolically active due to local northern climatic conditions and function rather slowly compared to ones in tropical latitudes [12].

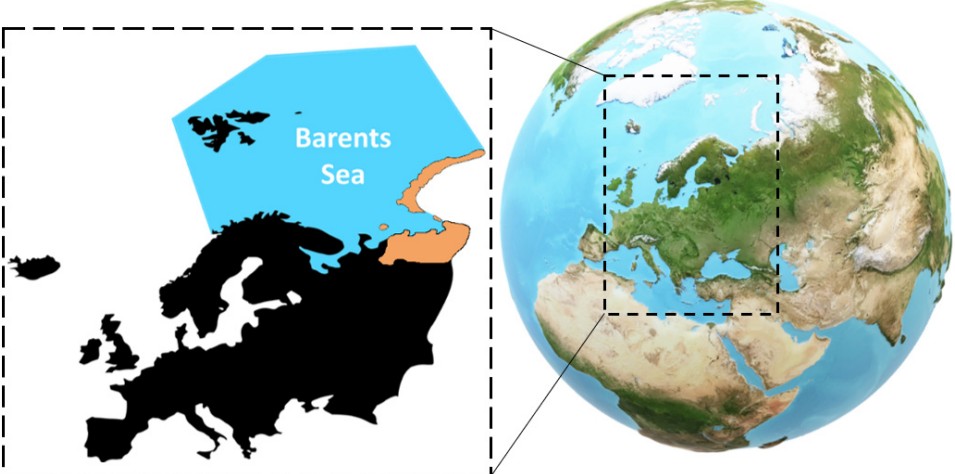

**Figure 1.** The Barents Sea: its location in relation to continental Europe and the study area in orange.

The polar marine climate of the Barents Sea favours long winters, short cold summers, strong winds in spring and autumn, and high relative humidity in the air. Overall, the weather is rapidly varying. Sudden changes in winds and precipitations are possible. Since the location of the sea is far north from the equator, the amount of incident solar heat is very low. The climate can be described as harsh and is characterised by extreme conditions of the Arctic [19]. In the perspective of potential oil spill response, this sea provides unfavourable prerequisites for natural oil slick photo- and biodegradation [25]. Due to its unstable water surface because of generated waves, the OSR operations may also be hindered.

2.1.1. Protected Areas

The Arctic is unique in terms of ecosystem functioning and vulnerability. Many land areas around the Arctic Ocean are ecologically sensitive and are on a list of protected areas. Only along the Russian Arctic coastline, there are about 35 nature reserves, which have the total area of about 39,000 km$^2$ [26]. The Barents Sea is not an exception from this either. In the north, on Svalbard archipelago, there are flora and fauna protective regulations with an emphasis on polar bear population protection. All touristic and domestic activities are limited in the area in order to protect its sensitive natural heritage [27,28]. In the west, Kola Peninsula has four of both terrestrial and nearshore protected areas along with several others inland. In the southern part of the sea, according to the World Database on Protected Areas, there are also several reserves [29]. On the mainland, there are five protected areas in proximity to the southern coast of the sea as seen in green in Figure 2: "Kolguevsky", "Nenetsky", "Haipudyrsky", "Yugorsky" and "Vaygach". As for fauna in these territories, there are bird habitats across the region but also various mammals. Among rare species, the representatives are such birds as cormorant, swan, sea eagle, sea ducks, and such mammals as grey lemming, wolverine and lynx to mention a few [24]. Many of them are on the list of the Russian Red Data Book and International Union for Conservation of Nature Red List of endangered species [28]. All these places have a special ecological emphasis on these local natural territories and demand a high level of OSR preparedness around them.

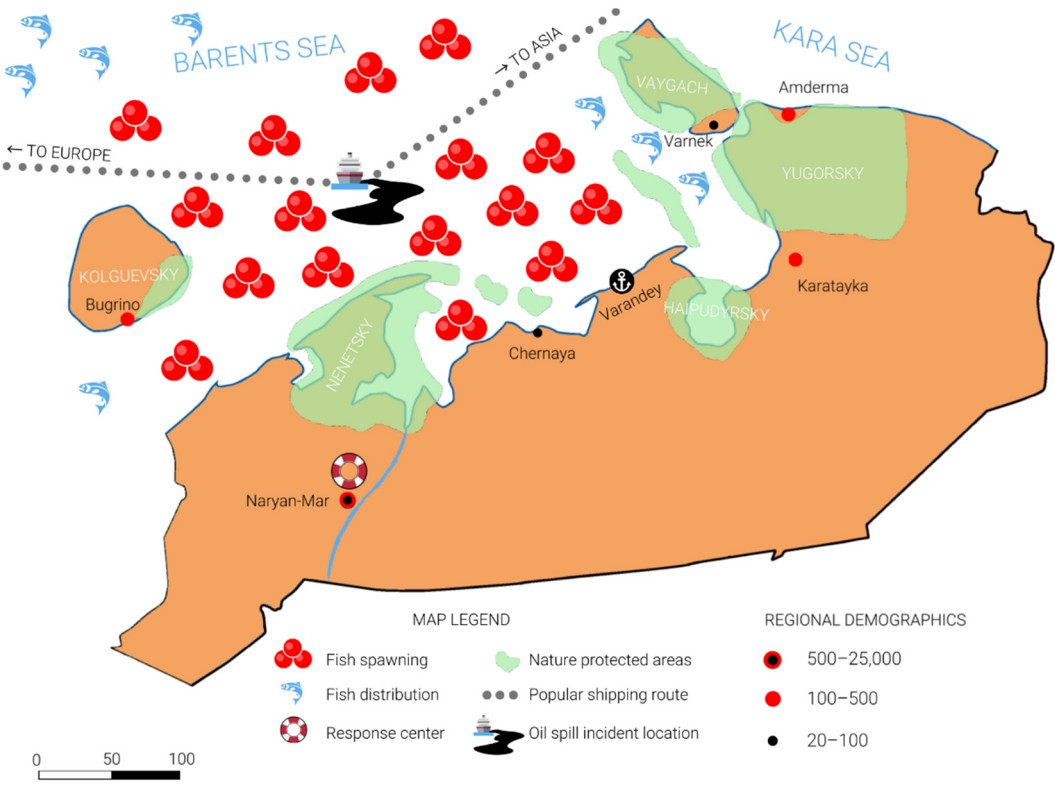

**Figure 2.** Nenets autonomous region in the hypothetical oil spill emergency.

2.1.2. Marine Resources

Regarding the abundance of marine ecosystems and fish species of the sea, such representatives as Greenland shark; Arctic skate; glacier lanternfish; deepwater redfish; sculpin (such species as Atlantic hooker, shorthorn, moustache, bigeye, polar, Arctic staghorn, twohorn, fourhorn, ribbed); lumpsucker (Atlantic spiny and leatherfin species); sea tadpole; eelpout (with representative species such as pale, polar, Arctic, threespot and longear eelpout); snakeblenny; Greenland halibut; Atlantic capelin; Atlantic and Polar cod; hamecon; Arctic alligatorfish; Atlantic poacher; lumpfish; stout and slender eelblenny, and daubed shanny—belong to the Barents Sea habitat. As an example, both polar cod and lumpfish have the same spawning and distribution areas in the sea, as shown in Figure 2 [30]. The only difference is in the spawning season: for polar cod, it is December to March, whereas for lumpfish—March to May [31,32]. For some fish species, the Barents Sea serves as a prevalent feeding area before spawning. Atlantic cod, being one of them, May to January, is often found here by commercial and local fishing vessels. In 2019, the cod stock was registered at about 1.5 million tonnes—to show a level of scale of its population [33]. Beluga, walrus, polar bear and white whale, for instance, being representatives of marine mammals, also have their habitats in the sea [24,34].

*2.2. Economic Activities*

From an economic perspective, the Arctic is an actively diversified region. Examples of activities present in the region may be fisheries and the oil and gas industry. Recent economic developments also include aquaculture, tourism, mining activities and sea transportation of oil products. By 2050, according to Corbett et al. [35], at least up to 5% increase of total shipping activities along the Arctic passages is expected. The main driving forces for it are global resource demand, growing population and availability of new maritime transport routes. In 2012, there were 1347 vessels registered in the Arctic region via the Automatic Identification System ship tracking. One third of them was carrying heavy fuel oil. The others were bulk carriers, fishing vessels, passenger cruises, general cargo carriers,

oil tankers and chemical tankers. Among them, larger ships operating in the area have to comply with the Polar Code and have certain ice class for crossing the Arctic Seas [36]. As regards to the recent data, 1628 vessels entered the Arctic area in 2019 [37]. This is about 20% higher than in 2012.

### 2.2.1. Marine Resources

Historically, the Barents Sea has been a reliable supplier of such benthic species as cod, haddock, halibut, and also other fish species such as polar cod, herring, mackerel and capelin [30,38]. This is not surprising since, as mentioned above, the sea is very biologically productive. In 2016, for instance, Norway reported record statistics of whole seafood export—2.6 million tonnes. The main importer is the European Union, buying about 70% of Norwegian seafood. The total value of 2017 selling was 9.8 billion euros. In case of the Barents Sea, there always have been certain quotas for the annual fish catch, binding both Norway and Russia, which mutually share the sea. This is a good practice supporting the idea of sustainable fisheries management, which has led to the local fish stock being one of the healthiest in the world. For 2017, in particular, the quota for cod catch in the Barents Sea was set at about 900,000 t, and for 2018—775,000 t [39]. It has approximately been the same for the last several years [40–43]. For 2020, the fish catch limit was recommended at about 690,000 t [33]. In the case of Russia, and its Murmansk sea port in the Kola Bay, being the largest above the Polar Circle, and the only ice-free port in the country—it also relies extensively on fisheries originating from the Barents Sea. The fish plays an important role in food supply in the region and beyond its borders: e.g., St. Petersburg area [25,44,45].

### 2.2.2. Transportation

Apart from fisheries, the Barents Sea is also considered as a very important transport route: not only for oil transportation, but also other goods. Regarding oil carriage, about 15 million tonnes were carried along the sea only within the Russian export scale during 2004–2009. The largest volume of oil transported was registered in 2010. The oil was mainly coming from an oil field developed in the Pechora Sea—the southeastern part of the Barents Sea, and counted for 15 million tonnes of crude oil [44]. In 2013, there was a start of "Prirazlomnaya" oil deposit development also in the Pechora Sea, and the first oil tanker shipping took place with its arrival to Rotterdam port in the EU. The annual deposit capacity to be reached by 2021 is 6.5 million tonnes of oil. As the size of specially built tankers "Mikhail Ulyanov" and "Kirill Lavrov" is 70,000 tonnes, then the calculated yearly number of round trips from the EU to the deposit would be 93 [46]. As a part of the Northern Sea Route, the Kara Sea has the key importance. Due to large developments in the mining industry and oil and gas sector, there is also ongoing export through the Barents Sea of gas condensate and multiple products of "Norilsk Nickel", which mainly represent non-ferrous metals: palladium, nickel, platinum, cobalt, copper and others [44,47]. In general, there is a number of terminals located in the Barents Sea, which make it busy in terms of maritime traffic [48]. This fact can be visually observed with a live traffic density map on a ship tracking website [49].

### 2.2.3. Oil and Gas Industry

The third important economic activity of the Barents Sea after fisheries and transport is energy resource deposits. Together with the Kara Sea, the sea contains about 70% of hydrocarbon resources of the continental shelf [11]. Norwegian surveys claim that the Barents Sea only holds about 2.8 billion m$^3$ of oil equivalent [50]. Consequently, it can be foreseen that the Barents Sea is being explored and will be explored in the upcoming years. In 2017, Norway, for instance, announced that 93 blocks were available for oil companies for license rounds [51]. Previously, oil and gas were explored strictly by the shore and within shallow waters—up to 70 m. Nowadays, fields of up to 700 m are considered for resource development [52].

### 2.3. Local Communities

One of the representatives of inland local communities is people of the Nenets autonomous region of the Russian Federation. It is the most sparsely populated region of the country, with a population of approximately 44,000 people. It is also the least populated area in geographical Europe. The population density is 0.25 people per square kilometre. This is about eight times less than in the most sparsely populated region of the least populated country in the EU—Lapland of Finland, and about five times less than the least populated state of the USA—Alaska [53,54]. As Figure 2 shows, there are several settlements of different population sizes by the seacoast of Nenets: Bugrino, populated with 417 inhabitants, Chernaya—with 22, Karatayka—with 544, Varnek—with 101 and Amderma—with 577. The capital of the region with 25,000 people—Naryan-Mar, is marked to demonstrate remoteness of the settlements from the regional centre, which also has the closest OSR centre and required equipment for operations at sea [55,56].

### 2.3.1. Indigenous Peoples

As overall regional statistics, the largest share of population in Nenets is demographically Russian—63%. The second and third largest are Nenets and Komi peoples, respectively 18% and 9%. Although Komi is only 9% of the population and it is twice less than Nenets, their population is about 350,000 people and they are widespread in the whole country around its various regions. Therefore, they are not considered in the section as indigenous for this particular study area. Nenets people, in turn, are known for their traditional reindeer herding, hunting and fishing in the region [57]. Reindeer herding is the dominant activity [58]. There are about 1000 people involved in it, managing more than 135,000 reindeers over 12 million ha. All reindeers are divided between 18 reindeer cooperative societies and are distributed across the region. Several of them are present around Chernaya, Karatayka, Varnek and Amderma [59]. In winter, reindeers are usually taken inland, and in all the other seasons, to the sea coastal areas. This takes place everywhere along the regional coast [60]. Indigenous hunting is mostly done for hoof and fur animals. Animal husbandry and cell breeding are also common these days. Fish catch is mostly traditional rather than commercial and related mostly to species of cisco and salmon. Salmon is represented by ten species, with its migration area in the study site. Among other popular fish species are navaga, Polar cod and Pacific herring, which feed on local phyto- and zooplankton. The spawning for navaga is from December to January, and for Polar cod—December to March. Polar cod is also an important link in the trophic chain, being a food for other fish, mammals and birds [24]. As an example of proper establishment of fishing, Bugrino has a fish cooperative society [61]. In total, there are 22 traditional cooperative societies in Nenets autonomous region. The remaining four deal specifically with fishing. Annually, October to March, up to 500 local fishermen are winter fishing at sea. Besides the previously mentioned species, they also catch Arctic flounder and Pacific rainbow smelt. In addition to indigenous traditional activities, there are also certain places of cultural heritage present in the region, where spiritual traditions are practiced [55,62]. Many of sacred places are found on the island, where Varnek is located—Vaygach island [63].

### 2.3.2. Regional Development

The region is part of the Arctic and has severe northern climatic conditions. This inhibits its economic development, primarily due to infrastructure insufficiencies. However, there are a number of local industries which are established in the area. One of the most developed economic sectors is fossil fuels extraction: both oil and gas, and mineral resources [57]. The share of both is the largest, when comparing with other sectors of the economy. Due to the presence of the onshore and offshore oil and gas industry, an oil terminal was opened in an abandoned regional settlement—Varandey. Nowadays, there are about 200 people living there. Apart from a couple of regular families, it is mostly inhabited by industrial workers, operating for oil and gas industry purposes locally in the

region [64]. In Nenets, Varandey is the only seaport, except for Amderma and Naryan-Mar down the river. Varandey terminal is marked in Figure 2, where oil is usually transported from to Murmansk, then further on across Scandinavia and towards the EU centre [55]. Besides the oil and gas industry, there is also a noticeable operation of such sectors such as tourism and traditional activities. As mentioned, it is primarily reindeer herding and fishing. The region is seen as a cultural touristic attraction because of expressed indigenous traditions and lifestyle. Similar cultural sights might be seen in Finnish Lapland. Since the territory is remote and pristine with several nature reserves, ecotourism is also developing in Nenets: e.g., in "Kolguevsky", "Nenetsky" and "Vaygach" protected areas. All these were mentioned in the development strategy of the region by 2030 and on the official Nenets website [60,65].

### 3. Location, Methods and Oil Characteristics

*3.1. Location*

The purpose of the work is to overlap several layers of data (environmental set of factors with oil spill-related probability map, proximity of response infrastructure as well as technical drawbacks of response technologies) into one discussion of hypothetical risks of oil spill accident in the Barents Sea to the environment. The selected geographical location of the oil spill, having the coordinates at 69.49° N 53.42° E, is near Novaya Zemlya and in close proximity to the Kara Strait. The choice of that particular study spot is explained by several reasons. Firstly, it lies along one of the most popular maritime passages in the local area, which makes it lively in terms of traffic activities [65]. This has been confirmed, for example, by Arctic shipping status reports and online marine traffic density maps— showing the scale of several hundred ships a year in the area [49,66,67]. Secondly, the selected regional spot is also characterised by local oil and gas activities. There are several offshore oil deposits northeast from Chernaya settlement. Despite the presence of the oil and gas industry in Nenets with its oil terminal in Varandey, West-East and East-West transportation routes are mostly transit without stops on the mainland [60]. Thirdly, it has a large economic activity besides the oil and gas industry. Several sources confirm the minor presence of commercial fisheries in the study area: mostly from September to December on an annual basis [30,60,68]. Among the examples may be navaga, Polar cod and Pacific herring [24]. Fourthly, there is also rich natural heritage and large biological diversity in the local environment including, for example, harp seals, polar bears, walruses and belugas. Out of all marine fish species of the Arctic Ocean, the Barents Sea hosts approximately one fourth of its overall stock [38,69].

The location of hypothetical accidents was intentionally chosen in the remote site. It is placed far from the regional search and rescue OSR centres, which are located at Murmansk, Archangelsk and Vorkuta. The closest OSR centre, Naryan-Mar, is 250 km away from the accident site (Figure 2). The distance from the spill to the nearest nature reserve in the south—"Nenetsky zapovednik"—is 70 km.

*3.2. Methods*

Oil drift simulations were performed with OpenDrift (Release v. 1.2.0), an open-source Lagrangian framework developed in Python by the Norwegian Meteorological Institute [70]. It is a programme used to simulate movement of objects in ocean and atmosphere. The current work required only oceanographic application. OpenDrift contains sub-modules that simulate the trajectory of different floating objects in the ocean, ranging from search and rescue to oil drift. Among possible examples of modelled trajectories may be such objects as microplastics, drifting vessels, boats, bodies and many others. For oil spill drift trajectory modelling, there is a specialised module—OpenOil. It has been used in this study to give realistic results: as, for example, with Deepwater Horizon oil drift simulations [71]. OpenOil simulates oil spill transformation, and also tracks oil changes in time and space. The slick pathway is visualised as a trajectory of individual dots in movement. Based on the generated visual picture, it is possible to make relevant general

conclusions about mitigation tools and clean up strategies as well as environmental or economic risks in the polluted areas. OpenDrift works with input oceanographic and atmospheric data: in our case, we used ERA Interim reanalysis and SVIM. OpenOil is configured with such physical processes as natural dispersion, evaporation, emulsification and biodegradation as weathering processes, using Euler interpolation scheme and a certain time step. The oil spills are simulated using 3D module, with vertical mixing due to oceanic waves and currents, resurfacing of oil because of buoyancy—thus taking into consideration oil physical properties. All these help to study a range of oils rather than concentrating only on one particular type.

Our experiments were designed based on the World Meteorological Organization's suggestion that a climate's normal periods are usually 30 years. Hence, the duration of each study set was also three decade long. Due to its availability, past weather data was used: from 1988 to 2018. Four seasons of the year were studied to observe seasonal differences and produce probability maps for each. Seasonal month settings were as commonly used: winter—December to February, and so on. The source point for all simulations stayed the same. The experiments were run with a 1-hour time step. Oil spill was released daily on a continuous basis, 100 oil particles per day during the selected study period (30 years). Two oil types were tested to generate the probability maps presented in Figure 3. Here, oil particles are considered stranded once they touch/are in nearest proximity to the coastal black line. In the produced maps, the remaining oil mass particles will reduce mass due to evaporation, and physical influence of sea such as waving activity causing dispersion of oil and other processes. Oil particles with remaining mass by the end of simulation will be weeks to months old. Thus, the particles in Figure 3 mainly illustrate the most likely oil transport from the source point marked with a red star. In real life, such continuous oil spills are unrealistic. Neither well blow-out, tanker accident nor pipe leakage can cause spill for such a stretched period of time. However, these experiments were made to offer statistically relevant data for the Barents Sea: footprint of oil contamination associated with the potential spill. They provided the necessary scenarios of oil drifting trajectories under local dynamic oceanographic conditions to demonstrate the most probable pathways of oil transport at sea surface and to envision coastal locations at risk of oil contamination.

In addition to the modelling tool, two publicly available online tools were applied for the study region: circumpolar oil spill response viability analysis tool and the Arctic risk map [72,73]. Analytical and synthetic methods were also used in the research: theoretical analysis and synthesis of various literary sources. These included local economic practices, strategies of regional development, its implementation plans, national legislation and regulations, publicly available materials from government websites, multiple academic journal articles and other relevant publications.

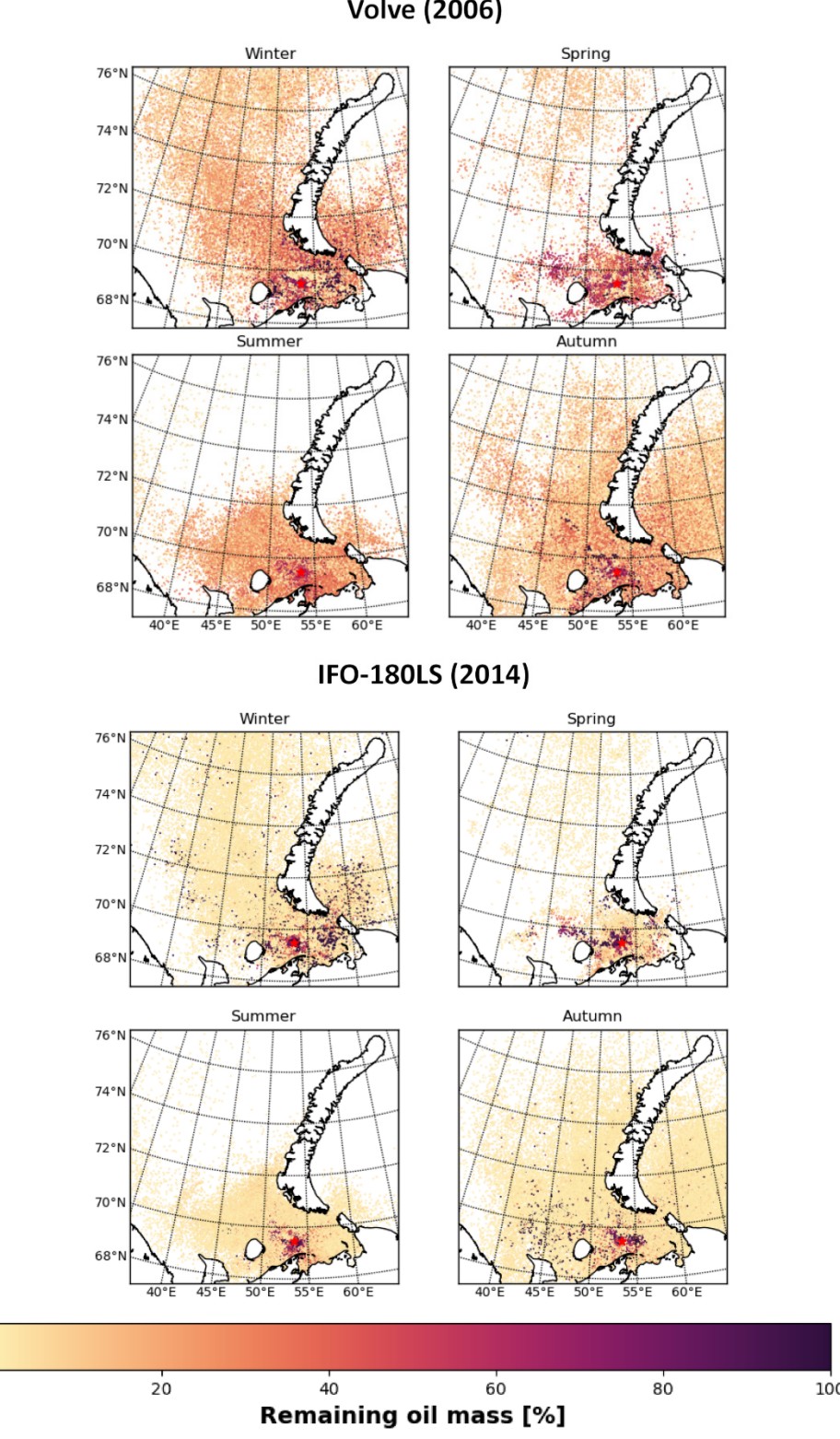

**Figure 3.** OpenDrift simulations for two studied continuous surface oil spills in the Barents Sea for annual seasons of 1988–2018: Volve (2006) and IFO-180LS (2014). Each presented probability map illustrates the initial spill location (marked with the red star) and seasonal patterns of remaining oil mass of each particle coloured by percentage relative to initial mass (shades of purple, as the colour palette).

*3.3. Oil Characteristics*

　　The simulations of oil spill accidents in four annual seasons included two oil types. One was a traditional heavy fuel oil–IFO-180 LS (2014), which is likely used by ships crossing the study area [74]. The Arctic risk map shows that such class of residual fuel oil is used in the area [72]. Volve (2006) was the other one, which is characterised as a medium oil type. It was additionally selected to give a broader picture when thinking about oil classes which might be transported from Europe and Asia, and to intercompare two different oil types in terms of their sea surface spreading. Regionally developed Arctic oils are mostly from light oil class, despite their higher viscosity. They are well studied because of local oil and gas industrial developments [75]. They also seem to be less impactful for biological species: from sensitivity and vulnerability perspectives [76]. Hence, this class was not chosen in our research.

　　OpenOil is based on a library of 1000 oils developed by the National Oceanic and Atmospheric Administration (NOAA). The library includes all necessary properties of various weathered oil types, showing changes in their physical and chemical state in time. Table 1 illustrates classification proposed by the World Petroleum Conference [77,78], with ranges of API gravity and oil density in relation to each category.

**Table 1.** Oil classification: according to its API gravity and density.

| Oil Category | API, ° | Oil Density ($\rho$), kg/m$^3$ |
|---|---|---|
| Light | API $\geq$ 31.1 | $\rho \leq$ 870 |
| Medium | 22.3 $\leq$ API < 31.1 | 870 > $\rho \geq$ 920 |
| Heavy | 10 $\leq$ API < 22.3 | 920 > $\rho \geq$ 1000 |
| Extra-heavy | API < 10 | $\rho$ > 1000 |

　　API gravity (°) for the chosen oils can be calculated as follows [77]:

$$API = 141.5d\,(60/60) - 131.5, \tag{1}$$

where d is the relative density of the oil (kg/m$^3$) and 60/60 °F is the specific gravity of the oil at 60 °F compared with that of water at 60 °F.

　　The respective densities and API gravities of the oils were: IFO-180LS (2014) of 975.8 kg/m$^3$ and API $\cong$ 13.5°, and Volve (2006) of 892 kg/m$^3$ and API $\cong$ 27.13°. Distinct classes of oil, as the ones considered in this research, are differently impacted by weathering processes. Evaporation, for instance, affects more lighter refined oil products than heavy crude oil products since the former presents higher concentration of volatile components than the latter [79]. Therefore, oils can prevail in the environment for longer or shorter periods depending on their type and it is thus possible to evaluate which one could impact more severely a given region. The presence of sea ice adds new complexities to the oil modelling task. The ice cover dampens wave activities, thus decreasing dispersion, entrainment and emulsification [80]. Oil-in-ice studies conducted since around 1970 have shown that the oil surface drift is highly dependent on the sea ice concentration, C. As a rule-of-thumb, oil tends to drift as in open waters when C < 30% and with the ice field for high sea ice concentration values (C > 80%). No general consensus is yet established for the intermediate range. Nordam et al. (2019), for instance, suggested a linear transition between the two extremes whereas French-McCay et al. considered that oil drifts with the ice field already for C > 30% [81,82]. Overall, it is observed that the sea ice limits the oil drift and spreading, thus restricting their displacement.

## 4. Results and Discussion

*4.1. Oil Behaviour and Spill Response Efficiency*

　　Figure 3 shows scenarios of oil drifting trajectories under local oceanographic conditions and demonstrates most probable pathways of oil transport at sea surface. It displays differences in intensity of remaining oil mass on the sea surface: from lowest values of

less than 20% to highest—at times more than 80%. The colour intensity shows how much oil likely remained on the surface: the darker the colour shade, the more oil remained on the surface. Findings from the conducted modelling showed that no distinctive elongated high-density oil trajectory patterns were observed in seasonal distribution of all simulations, which could occur, for example, in tropical regions caused by local oceanography. In contrast, oil had a rather even distribution of particles which tend to be carried in all directions from the initial spill location.

In a series of numerical experiments conducted in the Barents Sea by the RGC Risk Informatics, the Wild World Foundation showed that, in comparison to simulations performed during ice-free conditions, the area covered by the oil slick is more than 50% smaller when sea ice is present [83]. As can be noticed in Figure 3, particles corresponding to spring simulations were predominantly centred around the releasing point, similarly to the results presented in the aforementioned study. The smaller drift of particles in the spring simulations, which is also the period of maximum sea ice extent in south-east of the Barents Sea, might be seen as an indication of its role in the transport of oil.

### 4.1.1. Oil Types

The two selected scenarios, bunker oil IFO-180LS (2014) and medium oil Volve (2006), demonstrate that there is an observable difference between the two oil types. Volve has a more distinctive drifting pattern with wider surface coverage and potential to reach the nearby coasts of the archipelago, islands and mainland. IFO-180LS also has a wide polluted area, although less oil mass remained at sea. The share of weathering processes is noticeably higher. Both cases illustrated extensive trajectory movements and spill travel distances, at times exceeding 1000 km, when particles were found north of Novaya Zemlya. Pollution occurred in all water areas around the archipelago, except its north-eastern part. Small amounts reached Kola Peninsula in the west, 500 km away from the accident site. Some amount of the spilled oil drifted through the Kara Strait. The epicentre of the largest spill is within 150–200 km: with borders of the Kolguyev island in the west, Novaya Zemlya in the north, the Vaygach island in the east and the mainland in the south. Between two oil types, Volve has a much more noticeable presence in Figure 3 than IFO-180LS. This means more remaining oil mass presence, and, hence, medium oil types may be considered as a more hazardous oil type to the marine environment.

### 4.1.2. Summer

During summer months, most of the oil stayed within the radius of 300–400 km. This makes it a more localised emergency, which is a positive parameter from oil spill response preparedness perspective but a negative circumstance for the local marine and coastal ecosystems—it will be subjected to a higher oil concentration.

### 4.1.3. Autumn and Winter

In contrast, oceanographic and atmospheric conditions in autumn and winter seasons would result in a drift to the furthest distance in all directions. This may be explained by the presence of seasonal variability in sea and wind dynamics. A hypothetical oil spill accident may likely occur as a regional scale emergency—posing infrastructural difficulties for oil spill response operations. In both seasons, high oil particle density spots are also observed on the entire area of oil distribution. It is especially visible in the case of IFO-180LS (2014): dark spots in Figure 3, mostly west of the hypothetical spill accident site.

### 4.1.4. Spring

Spring simulations demonstrate that the highest oil remaining mass will concentrate in a small area producing most of the sea pollution in the land-surrounded region. This can be seen in the case of the Volve map where the remaining oil mass is mostly from the colour shade region of between 40 to 80% on the palette. Such oil distribution can provide a very noticeable impact within the accident area. The oil, however, is not localised similar

to the summer pattern but is still likely to travel farther north in the Barents Sea. Much of the oil particles are observed west of the archipelago.

When we overlap the generated probability maps with already existing local sensitivity maps to oil spills, we can see where environmental impacts can be observed the most [84]. Figure 4 illustrates sensitivity areas based on spaced distribution of such biological species as phyto-, zoo-, ichthyoplankton, zoobenthos, fish, mammals and seabirds.

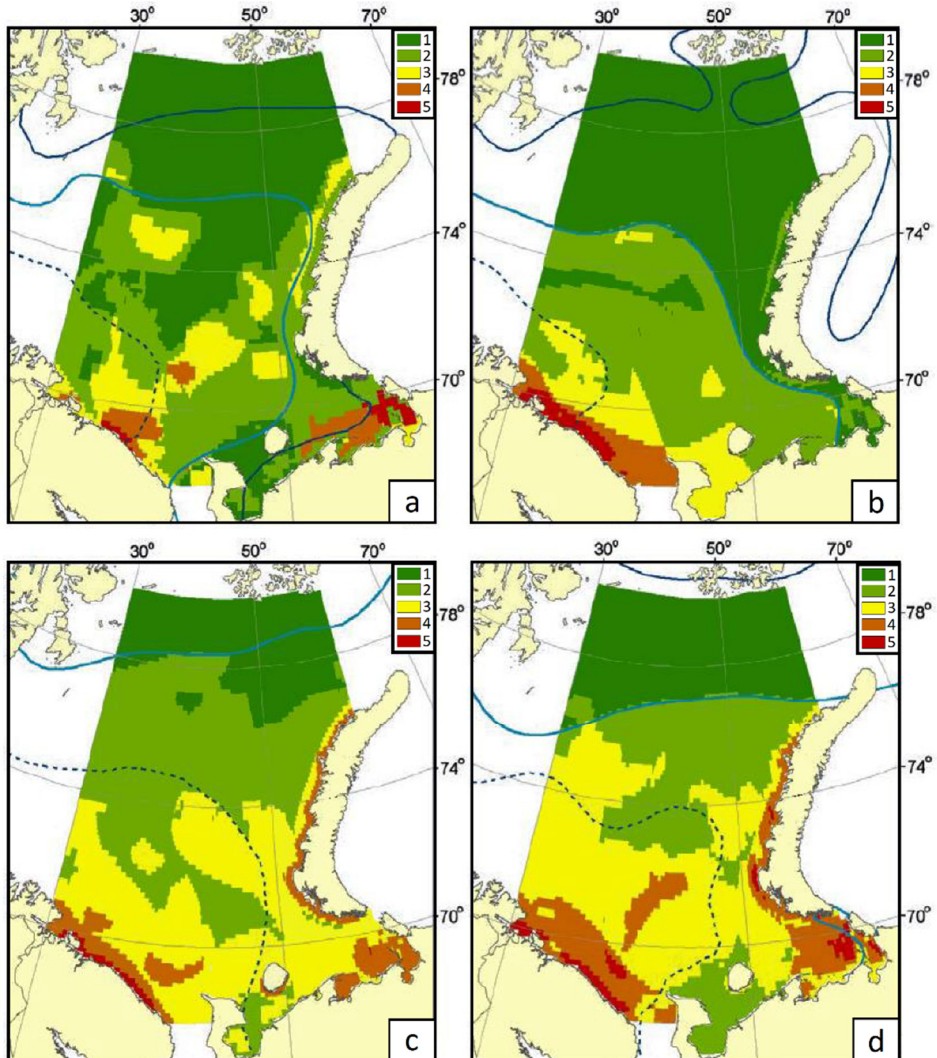

**Figure 4.** Sensitivity map of the Barents Sea: (**a**) Winter; (**b**) Spring; (**c**) Summer; (**d**) autumn [84]. Numbers in legend correspond to different sensitivity levels (1—very low, 2—low, 3—moderate, 4—high and 5—very high).

The presented data in Figure 4a for winter season shows higher susceptibility regions to the east from the initial accident site near Vaygach island. In case Figure 4b which is spring time, sensitivity levels near the epicentre are reasonably low: range of 1 to 2. Cases in Figure 4c,d, respectively for summer and autumn, illustrate similar trends: highly sensitive significant areas in the south-east of the sea and vast thin regions along the western coast of Novaya Zemlya: levels 4 and 5. Coastline of the Kola Peninsula has high susceptibility level in all four cases. All sensitive areas mostly correspond to seasonal species distribution [84].

Of the four seasons, the largest coverage for both studied oil types is observable during winter and autumn, whereas the coverage during spring and summer is smaller in size. In all seasons except summer, oil particles reached their highest latitudes of the Barents Sea, thus, drifting to remote areas of the high north. If hypothetical, summer oil spill accidents

may be likely called local, in all other seasons, it can take a regional scale. Based on the 30-year simulations presented here, the oil spill may likely drift much further than the location of the nearest oil spill response centre in Naryan-Mar, thus, putting extra attention to accident preparedness in the study region.

In order to define efficiency of oil spill response operations in case of an accident in the study area in all seasons, the Circumpolar Oil Spill Response Viability Analysis tool was used. It has three primary OSR technologies with a few subdivisions in each: mechanical recovery, dispersants' use and in situ burning. They are all listed in Table 2. Numbers show how favourable the study site is for oil spill response operations. Here, the amount of percent means response conditions are either favourable (marked with plus), marginal (marked with plus-minus) or unfavourable (marked with minus) for a certain tool throughout a certain season. For example, if the tool has 0% in the plus column, 56%—in plus-minus and 44%—in minus, it stands for zero favourable conditions in the area for the season, 56%—marginal and 44%—unfavourable. It is 44% unlikely and 56% marginally likely that the tool will be an applicable abatement method for oil spill response.

**Table 2.** Mean seasonal operability ("+"—favourable conditions, "+/−"—marginal conditions, "−"—unfavourable conditions) for oil spill response methods in the Arctic marine environment.

| OSR Tools | | Seasonal OSR Operability, % | | | | | | | | | | |
|---|---|---|---|---|---|---|---|---|---|---|---|---|
| | | Autumn | | | Winter | | | Spring | | | Summer | | |
| | | + | +/− | − | + | +/− | − | + | +/− | − | + | +/− | − |
| Dispersants | vessel application | 3 | 18 | 77 | 0 | 56 | 44 | 3 | 54 | 44 | 60 | 28 | 13 |
| | helicopter application | 19 | 68 | 14 | 5 | 5 | 90 | 19 | 21 | 61 | 54 | 19 | 27 |
| | aircraft application | 9 | 58 | 34 | 0 | 6 | 94 | 3 | 21 | 77 | 10 | 47 | 44 |
| Mechanical recovery | 3 vessels with boom | 20 | 75 | 6 | 0 | 6 | 94 | 2 | 10 | 89 | 19 | 25 | 56 |
| | 2 vessels with boom | 27 | 12 | 61 | 0 | 37 | 63 | 3 | 34 | 63 | 58 | 28 | 14 |
| | 1 vessel with outrigger | 2 | 31 | 68 | 0 | 22 | 78 | 0 | 14 | 85 | 37 | 44 | 19 |
| In situ burning | ignition from vessel | 5 | 55 | 40 | 0 | 26 | 74 | 2 | 27 | 71 | 20 | 56 | 24 |
| | ignition from helicopter | 5 | 7 | 88 | 1 | 2 | 97 | 4 | 13 | 84 | 11 | 21 | 69 |

As seen from the table, the selected geographical location has the following estimates for successful oil spill response. In autumn, four out of eight available Arctic OSR tools are unlikely (−) to be applied, whereas the other four have marginal (+/−) local operational performance in the season. Winter and spring seasons are unfavourable (−) for almost all OSR tools. Summer conditions are best for vessel or helicopter application of dispersants or two-vessel boom containment, which is marked with higher percentages in "+" column of the table. However, it is important to consider limitations of the search and rescue helicopter flight distance of about 175 nm [85]. If we discuss the local context of Russian legislation, dispersant use in the Barents Sea is also subject to special permission: its application is assessed according to net environmental benefit analysis procedure [86]. In situ burning opportunity is available in summer or autumn with about 50% chance for its operability in the local environment, but still is put in the marginal segment. As a general conclusion, during the year, overall favourable conditions for oil spill abatement are rather unlikely: 20 "−" cases, 9—"+/−", and 3—"+". This greatly relates to the local environmental conditions, which limit performance of the OSR methods. This includes extremely low temperatures, excessive ice cover, high wind and wave activity, daylight unavailability, precipitation storms and low visibility. As a representative example, here are climatic parameters above which no oil spill response is possible: air temperature (−18 °C), wind (15 m/s), wave height (4 m), light availability (darkness) and visibility (air: 4.0 km; water: 0.3 km) [4].

In order to reduce the likelihood and severity of oil spill accidents to the Arctic environment and local communities, we recommend that shipping routes are selected seasonally, taking into consideration sensitivity maps, and avoiding areas of high environmental vulnerability. For example, in the studied region, summer seems to be the most suitable for

shipping with the highest OSR operability and statistically least sea surface area polluted by a potential oil spill. However, oil spill preparedness should be as high as technologically possible, since summer has also several highly sensitive areas in the local marine environment. Here, most of the negative impacts will be localised within 300 km of the accident site—meaning exposure to higher oil concentrations. Out of four seasons, autumn seems to be one of the worst seasons for shipping due to mostly marginal/unfavourable oil spill response conditions and vast area of oil spill distribution. Winter, just as autumn, has a likely wide oil contamination area but also has unfavourable conditions for oil spill abatement. This season could be the second worst after autumn. The third worst is spring, having low chances for successful oil spill operations with existing OSR tools. As in cases of autumn and winter, a spring oil accident is likely to have regional scale.

This preliminary assessment aimed at drawing attention to the scale of hypothetical impact, and to provide a motivation for detailed risk assessment, in order to avoid possible environmental and socio-economic damages to this region of the Barents Sea. From our opinion, it is important to note that regional impacts and climate conditions similar to the 1988–2018 oceanographic and atmospheric data set may be expected in the near future, within the next 10 years. The short time span is due to possible unpredictability of future climate variations. As reported by IPCC in 2014 [87], the Earth's surface was successively warmer in each of the prior three decades than any preceding decade since 1850. Climate change is a self-accelerating process, especially in the Arctic latitudes. However, there is certain unpredictability of how fast and severe those changes may be and how it will affect oceanographic and atmospheric phenomena. If the climate normal of the last century was 30 years as a standard, this may change during the course of the 21st century. There is already discussion in the National Oceanic and Atmospheric Administration (NOAA) to reconsider the 85-year-old concept of climate normal, due to an accelerating trend of ongoing climate change. NOAA suggested to work with shorter periods of 20, 15, 10 and 5 years [88]. As an illustration to this, we can compare two recent periods in temperature anomalies—1980–2018 and 2008–2018, which are shown in Figure 5.

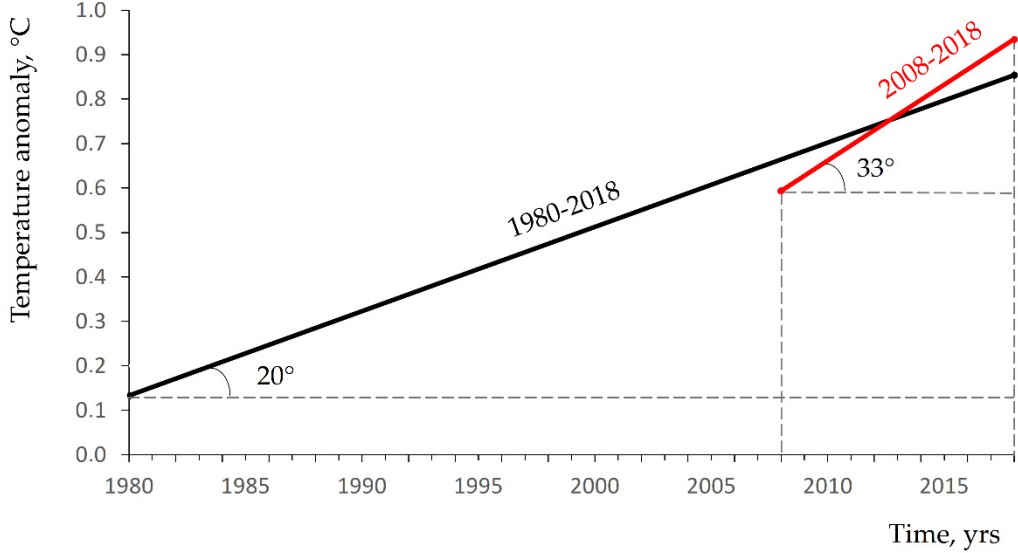

**Figure 5.** Acceleration of climate change: black line period of 1980–2018 and red line—2008–2018.

The trendlines illustrate the change in global surface temperatures relative to 1951–1980 average temperatures. The observation shows us that temperature rise accelerated in the last decade having a much more rapid tendency of deviation from climatic norm [89].

**5. Conclusions**

Simulations of surface oil spills were conducted in south-east of the Barents Sea to identify oil pollution trajectories. The study site is located near ecologically and economically important areas and lies along one of the most popular maritime routes. Two oil types were selected to demonstrate oil behaviour in the same Arctic environmental conditions in case of an oil spill: a popular bunker oil and a medium fuel oil. The study set provided statistical results for winter, spring, summer and autumn environmental conditions. Both cases demonstrated extensive spill travel distances, exceeding 1000 km and posing infrastructural and technological challenges for response operations. The selected geographical location has mostly unfavourable environmental conditions for oil spill abatement. Out of eight available Arctic oil spill response tools, only three showed potential for application in such climatic region and only in summer. Autumn, winter and spring seasons remain mostly difficult for organisation of abatement operations at sea. The area of highly probable impact is bordered by the Vaygach island and Novaya Zemlya in the east, the mainland of the Nenets autonomous region in the south, and the Kolguev Island in the west. In case an oil spill would occur at the location, all coastal areas within 300 km reach would be statistically vulnerable all year. From an environmental perspective, various ecosystem levels can be subjected to the risk of damage. The study area has vulnerable species during the whole year, but the autumn period seems to be the most dangerous. From a socio-economic perspective, impacts to both commercial and traditional fishing activities are likely to occur, however, this needs to be addressed in more detail in a separate research initiative.

Of the four seasons, the largest coverage for both studied oil types is observed during winter and autumn, whereas the coverage during spring and summer is smaller in size. In all seasons except summer, oil particles reached their highest latitudes of the Barents Sea, thus, drifting to remote areas of the high north. If hypothetical summer oil spill accidents may be likely called local, in all other seasons, it can take a regional scale. Hence, the accident consequences are most localised in summer in comparison with the other three seasons, where oil travel distances are large. Considering probable oil trajectories and possible impacts to the environment, summer would be the recommended shipping season along the studied route: e.g., oil and transportation of oil products. Autumn, in contrast, could be considered as a highly sensitive season for coastal offshore and onshore communities which are north from the initial spill site. Taking into account oil drift trajectories reaching the southern coast of Novaya Zemlya with its biodiversity levels, autumn shipping needs a higher level of oil spill response preparedness. The possibility of lowering the amount of shipping traffic in this season could be considered.

The produced probability maps indicated oil distribution patterns which may cause potential threats to local ecosystems. It is expected that the outcomes of this research are valuable to local and Arctic authorities, industrial actors and coastal communities. It may also contribute to creating a more extended picture for a hypothetical oil spill case in the considered local geographical point of the Barents Sea and improve local oil spill response preparedness. However, further research in environmental and socio-economic risk assessment is recommended to gain a detailed understanding of impacts to local flora and fauna as well as indigenous communities.

**Author Contributions:** Conceptualization, V.P.; methodology, V.P., V.C.M.d.A., L.R.H. and E.P.; software, V.C.M.d.A. and L.R.H.; validation, E.P.; data curation, V.C.M.d.A.; writing—original draft preparation, V.P. and V.C.M.d.A.; writing—review and editing, E.P.; visualization, V.P. and V.C.M.d.A.; supervision, L.R.H. and E.P. All authors have read and agreed to the published version of the manuscript.

**Funding:** This research was funded by Northern Periphery and Arctic Programme 2014–2020.

**Informed Consent Statement:** Not applicable.

**Acknowledgments:** The authors acknowledge the "Arctic Preparedness Platform for Oil Spill and other Environmental Accidents" (APP4SEA) project co-funded by the Northern Periphery and Arctic Programme 2014–2020. We also thank Henrikki Liimatainen for help with contribution to this work and Murmansk Marine Biological Institute of the Russian Academy of Sciences for providing sensitivity maps of the study region.

**Conflicts of Interest:** The authors declare no conflict of interest. The funders had no role in the design of the study; in the collection, analyses, or interpretation of data; in the writing of the manuscript, or in the decision to publish the results.

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
