# Peer review of "A 30-Year Probability Map for Oil Spill Trajectories in the Barents Sea to Assess Potential Environmental and Socio-Economic Threats"

_resources, doi:10.3390/resources11010001_

Round 1

Reviewer 1 Report

Dear authors,

I reviewed this paper with interest but found it needing a better introduction and a complete deletion (or re-framing) of section 4.1. I am providing an annotated .pdf file with comments, which can be summarised as follows:

1. The Introduction should compare and contrast the approach in this paper with similar models undertaken for the Eastern Mediterranean by the NEREIDs and Sea4All projects. Please, see the papers from Coppini et al. for the Sicily Straight and also Alves et al. (2014) Mar. Poll. Bulletin, Alves et al. (2015). Environ. Pollution and Alves et al. (2016). Scientific Reports. I would also have a look at the papers from Zodiatis et al. (2016. 2017) on oil spill modelling and add 1-2 paragraphs about the state of the art in these parts of the world. These papers were produced out of a EU projects that were paramount to understanding the risk of oil spills to the Eastern Mediterranean.

2. I disagree with the part focusing on the estimated impact of human activities on the Barents Sea when stating climate change as the main impacting phenomenon - to me, it is obvious that the main impact has to do with Russia and Arctic countries using ever-so-increasing numbers of ice breakers and larger ships on the Arctic region. The Barents Sea has been a shipping corridor for centuries, and surely deploying all sorts of icebreaking ships will keep the shipping lanes open, regardless of climate change??? Please, delete. You are being speculative.

3. Table 3 is very poor. We are all vulnerable to oil spills, if we are close to them. This part of the paper is speculative and assumes that oil spills will be gigantic and not confined/mitigated on the spot. Do the authors know how much oil and chemical releases are recorded in the UK North Sea, per platform, per year? Approximately 1.3 cubic meters per platform, in a universe of 200+ platforms, and in a water column of 80-90 metres on average, that will dissolve the pollution in minutes. Please, delete section 4.1 and Figure 4 as you would have to make a much more detailed analysis (and another paper) to justify this section.

The figures with the oil fate models are too small. They need to be enlarged in the final draft of this paper.

Author Response

Thank you very much for your comments helping to improve the quality of our paper. We have addressed all of them and made according changes in the article text. Please see the pdf-attachment with more details on it.

Reviewer 2 Report

The authors address oil-spill preparedness in case of extreme condition (arctic). It is a supposed oil-spill in the South of the Barents Sea. Response Center location is a tricky question along the Nenets community coasts. The paper is very well written and interesting to follow. We suggest publishing the paper after adding several details about the hypotheses made and the results presented. The paper has quasi no typos. We give here under remarks from the scientific and editorial points of view.

Scientific

A main unknown remains after paper reading. It is about the hypothesis on a continuous oil release. For the accident considered, is-it an oil release coming from a well, a tanker or a pipe ? The authors could detail the hypothesis made on the continuous release (line 332). For example, oil ice-trapped can be removed continuously during ice melting.

The number of ships given at the end of section 2.2 could depend on the different arctic classes.

A remark about the compensation funds could be given. Financial compensation is difficult to quantify for indigenous people and mostly for cultural and environmental heritages (end of section 2 and line 527). An accident coming from a well or a tanker leads to different compensation regimes.

The ice coverage could bear on oil drift, which could be detailed (end of section 3).

Extremophile organisms could be added in the second paragraph of section 4.1.

Comparing years 2008-2018 with years 1988-2018 could show the climate change acceleration (line 552).

A place of refuge study could be proposed on a coast blocking the north drift during autumn (studied route, line 601).

Editorial

 thous > 1000, line 111

 (60/60) is unclear in equation (1).

The probability is explained in the caption of figure 3 and it may be made in section 4.1.

Table 2 presents numbers, which could be clarified. First column of table 2: a separation line is missing between the two last items.

Edit sentence lines 488, 489 by indicating a risk map.

Typos

dot position, [14,15] . at line 74, idem lines 161, 266, 285

(.), ref 3

Mention the editors: refs 5, 10, 32, 36, 48, 73

Capital letter, John, ref 18

Author Response

(The authors gave the same response as above.)
